# A Method for Medical Microscopic Images’ Sharpness Evaluation Based on NSST and Variance by Combining Time and Frequency Domains

**DOI:** 10.3390/s22197607

**Published:** 2022-10-07

**Authors:** Xuecheng Wu, Houkui Zhou, Huimin Yu, Roland Hu, Guangqun Zhang, Junguo Hu, Tao He

**Affiliations:** 1School of Mathematics and Computer Science, Zhejiang A & F University, Hangzhou 311300, China; 2Zhejiang Provincial Key Laboratory of Forestry Intelligent Monitoring and Information Technology, Hangzhou 311300, China; 3College of Information Science & Electronic Engineering, Zhejiang University, Hangzhou 310027, China; 4State Key Laboratory of CAD & CG, Hangzhou 310027, China

**Keywords:** microscope image, autofocus, variance, NSST, sharpness

## Abstract

An algorithm for a sharpness evaluation of microscopic images based on non-subsampled shearlet wave transform (NSST) and variance is proposed in the present study for the purpose of improving the noise immunity and accuracy of a microscope’s image autofocus. First, images are decomposed with the NSST algorithm; then, the decomposed sub-band images are subjected to variance to obtain the energy of the sub-band coefficients; and finally, the evaluation value is obtained from the ratio of the energy of the high- and low-frequency sub-band coefficients. The experimental results show that the proposed algorithm delivers better noise immunity performance than other methods reviewed by this study while maintaining high sensitivity.

## 1. Introduction

As a complex and orderly system, life and its mysteries have always attracted the attention of scientists. Every major discovery of life is inseparable from technological innovation. As an important branch, microscopic imaging technology makes conducting life science research easier. Microscopic imaging technology was developed from the birth of microscope. In the late 17th century, Leeuwenhoek invented the microscope. However, the lenses produced at that time were relatively rough and the magnification was relatively simple. In the 18th century, Carl Zeiss began to manufacture composite microscopes. Due to the lack of scientific guidance, the optical quality of microscopes produced in the early stage was extremely unstable. At the end of 1860, Abbe cooperated with Zeiss to complete the design of the optical system, laying the foundation of the Abbe imaging principle, and thus, the theory of microscopic imaging. On this basis, microscope technology has experienced rapid development.

With the further development of medical technology and the transformation of the medical model, people are increasingly concerned about their own health, and the requirements for the quality of medical service are also increasing. The application of electron microscopy in the medical field, as a tool to study the microworld, has been widely used in many fields, opening up a new field of vision for medical research. This is true especially in virology, cell biology, histology, pathology, and molecular pathology. In the practical application of clinical medicine, electron microscopy plays an important role in the identification of disease conditions and causes, especially in the classification and diagnosis of tumors, kidney diseases, blood diseases, etc. [1] Both medical students, researchers, and medical personnel need to master the relevant knowledge of biomedical electron microscopy technology. In addition, urinary sediment examination is one of the most important means for medical diagnosis of human diseases, whether it is a routine physical examination or a patient examination at the time of medical treatment. Urine sediment contains red blood cells, white blood cells, crystals, sperm, epithelial cells, casts, and other impurities. The examination of urinary sediment has a very important reference role in the diagnosis and treatment of human kidney, bladder, and other diseases. If the urine contains a large number of white blood cells, it indicates that the patient may suffer from pyelitis, cystitis, or other diseases; if the urine contains a large number of red blood cells, it indicates that the patient may suffer from nephrotic nephritis, acute (chronic) glomerulonephritis, or other medical diseases. If epithelial cells appear in groups and leukocytes also increase, this indicates that tissues and organs with this type of epithelial cell suffer from inflammation; if calcium urate or oxalic acid crystals are detected in fresh urine, and more red blood cells are detected, the patient may suffer from urinary stone disease. When the urine contains more Gram-negative bacteria, the patient may be suffering from cystitis, pyelitis, or other diseases.

However, when the microscope imaging technology is used to obtain the image of urine sediment cells, the microscope platform always deviates from the best imaging position due to its own processing problems or if it is used for too long; thus, the image will become blurred, which directly affects the subsequent image processing and related research, and even causes researchers to misjudge the data. Therefore, in order to obtain a clearer and more accurate image of urine sediment cells, it may be necessary to choose a good autofocusing algorithm.

The innovation of this study lies in applying an improved NSST algorithm to autofocus the image of urinary sediment cells, and to obtain a clear image of urinary sediment cells. In order to verify the feasibility of the algorithm, different types of images are added for the experiments. The main contributions of this study are summarized as follows:The NSST algorithm can decompose the image at multiple scales to obtain a low-frequency sub-band and several high-frequency sub-bands, and the image information contained in different sub-bands is also different. By calculating the variance of different sub-band coefficients, the interference caused by the background of the urine sediment image can be further reduced and the performance of the NSST algorithm can be improved, as to obtain a better clarity evaluation curve.Microscopic imaging technology will inevitably generate noise in images due to factors such as environment, equipment, and improper operation. In order to simulate the noise situation as much as possible, different noises are added to the experimental image, and a bilateral filter and a Gaussian filter are applied to the noise image to improve the noise resistance of the algorithm. Finally, the noise resistance of the improved NSST algorithm and other algorithms in this study are tested under the same operating conditions, with the results showing that the improved NSST algorithm has a better anti-noise performance than other algorithms used in this study.

The organization of the remaining sections is as follows: The related work is presented in Section 2. In Section 3, the principle and motivation of the NSST algorithm are briefly introduced, some improvements to the algorithm are proposed to boost its performance, and some theoretical foundations of the NSST algorithm are explored, while implementation steps, flowcharts, and pseudo-codes of the algorithm will also be presented. In Section 4, the experimental results are discussed, including for the comparison experiment and noise immunity experiment. The last section is the conclusion of this study.

## 2. Related Work

To obtain sharp images by controlling a motor to adjust the lens or the stage in a computer-controlled microscope imaging process, autofocusing adopts two main methods [2]: active focusing, based on the principle of distance measurement, and passive focusing, based on image processing. Because microscopic images are often taken with a high-magnification and small-aperture objective lens, the depth of field stays at the micron level, and thus, the mechanical system is very demanding. In view of the defects of the active focus mode, the passive focus method based on image processing has been widely implemented, with the purpose of obtaining a real-time image dataset of the microscope, analyzing the current image focus state, and controlling the stepper motor to modify the step size according to specific search strategies and to obtain as sharp of an image as possible. As indicated by Qu et al. [3], even for the same sample, the performance of different focus criterion functions will be quite different. In this process, therefore, the key question is how to choose a suitable evaluation function to judge the sharpness of images.

In terms of the image sharpness evaluation, Xia et al. [4] have summarized and compared 16 traditional functions, including Laplace, Tenengrad, spatial frequency, wavelet transform, and fast Fourier transform, finding that Tenengrad performs best in the focus measure both for global search and local search, but has a weak noise immunity performance. In addition, Xia [5] proposed a new fusion algorithm by dividing information entropy and Tenengrad to enhance noise resistance, but it takes a bit longer timewise. Liu et al. [6] have designed a complex imaging process modeling and image sharpness evaluation method based on fuzzy entropy, a method that is robust to changes in noise, lens magnification, and illumination conditions, but which has a slow test speed, relatively fixed fuzzy control results, and poor flexibility. Alma Rocío et al. [7] offered a new autofocus and fusion algorithm (AFA) by selecting the best focus image (BFI) at different distances from a target in a bunch of images, and experimental results demonstrate that this algorithm is different from traditional fusion ones. For example, the AFA can boost image quality in a shorter time than traditional fusion methods, but its anti-interference performance in terms of noise is unknown. In order to accurately identify the contours of cells and count them, Hore Sirshendu et al. [8] proposed a novel method to develop the automatic qualification of cells’ contour identification, which necessitates marking cells. Thus, a novel method based on a new two 4 × 4 kernel for edge detection after the pre-processing step is employed. The proposed method was implemented and evaluated on light microscope images of rats’ hippocampi, as a sample of the brain cells. A comparative study for the proposed method was performed with other edge detection techniques, such as the Canny, Roberts, Prewitt, and Sobel protocols. The experimental results proved that the proposed method is superior to the other edge detection methods in terms of the accuracy, specificity, and false alarm count. Akiyama et al. [9] proposed a sharpness evaluation function based on Daubechies wavelet transform to deal with the high-pass filtering effect; this function has been used in ground vehicle infrared image guidance systems, but it has an impact on the accuracy. Based on wavelets for blood smears under a microscope, Makkapati et al. [10] further proposed an improved image autofocusing method, which can obtain a smooth sharpness evaluation curve without any fluctuations, thus improving the autofocusing accuracy; however, its focusing speed is reduced. Through machine learning, Chen and Peter van Beek [11] have designed a supervised machine learning method by using two decision-tree classifiers to control the focusing process and find the clearest image, but also at the expense of some focusing speed. For Depth from Defocus (DFD) estimation, S. Matsui and Taniguchi RI [12] proposed a new imaging technology, a half-scan imaging technology and a processing method, which has a high image signal-to-noise ratio, flexible adaptation to scene depth, and full compatibility with conventional imaging, among other advantages. However, although easy to implement in cameras, it is still not widely deployed in other scenes. By removing a large number of unnecessary image sub-blocks from the adjusting microscope image thresholds, Lv and Wang [13] determined the final focus window according to the sum of the gradient amplitude of the image sub-blocks, finally calculating the gradient and variance of the window sub-blocks while taking the pixel weighting method as the focus evaluation function. With noise added, it showcases good noise resistance and stability, but whether it can be applied to non-microscope images is still unknown. To improve the denoising ability of the algorithm, Samsad Beagum et al. [14] proposed an automation technology LPA-ICI-PSO by studying the local polynomial approximation (LPA) filter supported by the intersection confidence interval (ICI) rule (LPA-ICI) and combining it with particle swarm optimization (PSO). This guarantees less computational time along with optimal denoising compared to the LPA-ICI as established by the performance metrics. The experimental results established the superiority of the proposed LPA-ICI-PSO over the classical LPA-ICI filter. In the same year, Amira S. Ashour et al. [15] proposed a new automation technology, “standard optimized LPA-ICI” (SO-LPA-ICI), based on LPA-ICI-PSO. The denoising effects of different methods on rats’ renal microscopic images in the presence of Poisson noise were compared. The results showed that SO-LPA-ICI achieved fast denoising compared with LPA-ICI-PSO. Hamid Reza Shahdoosti and Omid Khayat [16] proposed an image denoising method which uses sparse unmixing by variable splitting and an augmented Lagrangian (SUnSAL) classifier in the non-subsampled shearlet transform (NSST) domain. The experiments demonstrate that the proposed approach improves the image quality in terms of both subjective and objective inspections, compared with some other state-of-the-art denoising techniques. Based on the above literature review, in order to improve the autofocus recognition of urinary sediment cells, this study proposes a sharpness evaluation function based on the non-subsampled shearlet transform (NSST). On this basis, the performance of the algorithm is further improved by a bilateral filter [17] and variance. The experimental results show that this method is effective and the interference of noise to images can be appropriately curbed. The ratio of high- and low-frequency energy coefficients is calculated by combining both frequency and time domains. Calculated with real values of coefficients, it has a lower complexity [18]. Theoretical analysis of the algorithm shows that the method may have obvious single peaks, a high accuracy, and few local peaks while delivering high algorithm noise immunity and focusing speed.

The datasets generated during and/or analyzed during the current study are available in the figshare repository (https://doi.org/10.6084/m9.figshare.19524283 accessed on 6 April 2022).

## 3. Materials and Methods

### 3.1. Non-Subsampled Shearlet Wave Transform (NSST)

The NSST [19,20,21,22] includes two parts: multiscale decomposition and multidirectional decomposition. In fact, the NSST is an optimized and improved version of the NSCT algorithm proposed by Cunha et al. [23,24,25], with the characteristics of multiscale, multidirectional, and translation-invariant transformation, as well as high-computational efficiency. The NSST can capture the geometric and mathematical properties of an image, such as scales, directionality, elongated shapes, and oscillations. The NSST is an optimal transform with sparse coefficients, in which a Parseval frame of well-localized waveforms at various scales and directions is formed. During the decomposition of an image, the NSST has no limitation on the number of directions for shearing [26]. After an image is decomposed with k-level non-subsampled pyramid (NSP) multiscale decomposition, k + 1 sub-band images can be obtained, including one low-frequency image and k high-frequency images with different scales. After multiscale decomposition, the sub-band image is decomposed in one-level multiple directions. NSST direction decomposition adopts a shear filter (SF) [27] to ensure that the images are not distorted, but provide translation invariance while effectively suppressing the pseudo-Gibbs effect. Moreover, the computational burden of the NSST is lower than that of the NSCT [26]. In addition, directional selectivity in the NSCT is artificially made by the special sampling rule of filter banks which often produces artifacts, whereas this defect has been resolved in the NSST. The NSST decomposition structure diagram is shown in Figure 1. Therefore, the NSST is selected in this study for a sharpness evaluation of medical microscopic images and makes appropriate improvements.

### 3.2. Analyses of Algorithm Theories

In the microscopic field of vision, the cell distribution in images is scarce and discrete, and the gray level of an image is similar. The cell image in the defocused state can be approximately regarded as a uniformly distributed circular light spot, and thus, the background would greatly influence the judgment of clarity. The low- and high-frequency information after NSST decomposition has different physical meanings. The coefficient with a large absolute value in the high-frequency information corresponds to the detailed information, such as edge, which is very sensitive to changes in image sharpness. If an image is mixed with noise, it will be mixed in with the high-frequency information after decomposition. In contrast, the low-frequency information can retain the general information of the original image and is not very sensitive to changes in image sharpness. Therefore, the characteristics of low-frequency sub-band coefficients can be used to improve the noise resistance of the evaluation function. In addition, the variance processing of the decomposed sub-bands after NSST decomposition can further diminish the impact of noise.

### 3.3. Algorithm Improvement

The variance function [28] represents the degree of dispersion of the image’s gray distribution. For out-of-focus images, the gray value transformation range is small, the degree of dispersion is low, and the variance is small; for focused images, the opposite is true. This focus measurement computes variations of pixel intensities and uses the power function to amplify larger differences from the mean image intensity, as defined below, where MN is the total number of pixels, g(i,j) is a pixel value, and $\bar{g}$ is the average pixel of the image.
(1)$$V=\frac{1}{MN}\underset{i,j}{\sum }{\left[g\left(i,j\right)-\bar{g}\left(i,j\right)\right]}^{2}$$

The sub-band coefficient energy is obtained by calculating the variance of sub-band images in different frequency bands and directions acquired in NSST decomposition. This can better extract the features of the image and improve the accuracy of autofocusing.

### 3.4. Algorithm Implementation

Based on the above discussions, certain implementation steps of the image sharpness evaluation algorithm combining NSST decomposition and variance processing are proposed as follows:
Perform NSST decomposition on an image to obtain one low-frequency sub-band and several high-frequency sub-bands.Obtain the variance processing coefficient $V_{L}$ of the low-frequency sub-band image in the NSST transform domain, as defined below:(2)$$E^{L}=V_{L}$$


Then, calculate the variance processing coefficient $V_{k,l}^{\prime }$ at each level of the high-frequency sub-band image in each direction, and add $V_{k,l}^{\prime }$ in different directions of the same frequency band to obtain the variance processing coefficients $SV_{k,l}^{\prime }$ at each level of the high-frequency sub-band image, as defined below:(3)$$SV_{k,l}^{'}=\underset{k}{\sum }\underset{l}{\sum }V_{k,l}^{'}$$

The image energy at each level of high-frequency sub-band $E_{k}^{H}$ is defined as follows:(4)$$E_{k}^{H}=\frac{1}{2^{k}}\sum SV_{k,l}^{'}$$

The weighted total energy of high-frequency sub-band images is defined as follows:(5)$$E^{H}=sE_{1}^{H}+\left(1-s\right)\frac{\sum_{k=2}^{N}E_{k}^{H}}{w}$$

3.Combine the energy of the high- and low-frequency components to calculate the sharpness evaluation value, as defined below:(6)$$h=E^{H}∕E^{L}$$

In this experiment, the parameters are w = 3 and s = 0.8. The specific flow chart of the experiment and the pseudo-code of the algorithm are shown in Figure 2 and Algorithm 1, respectively. In practical application, due to many reasons such as image acquisition equipment and natural environmental factors, the processed image is different from the real image. In order to simulate this difference, we add noise to test the noise immunity performance of the algorithm. As to how to add noise, see the experimental results in Chapter 3. SNVRO is the operation flow without adding noise.
**Algorithm 1. Pseudo-Code of the Algorithm****Input**: *N*_1_ is the number of pictures to be processed.
**Output**: *H* is the definition of the evaluation value.
1:L←12:N←3 Decomposition series3:**for**L to $N_{1}$ by 1 **do**4:  im←imread(‘image path’)5:  im←rgb2gray(im)6:  im←im2double(im)7: The image is decomposed by NSST to obtain low-frequency component $f_{1}$ and high-frequency components $f_{2k},f_{3k},\;\mathrm{and}\;f_{4k}$.8:  $\left[V_{1},V_{2},V_{3},V_{4}\right]\leftarrow get\_variance\left(im,f_{1},f_{2k},f_{3k},f_{4k}\right)$9: The energy coefficients of each high-frequency sub-band are processed and added to obtain $E_{1},E_{2},\;\mathrm{and}\;E_{3}$. The low-frequency coefficient is $E_{L}$.10:  $E_{H}\leftarrow s*E_{2}+\left(1-s\right)*\left(E_{3}+E_{4}\right)/N$11:  $H\leftarrow E_{H}/E_{L}$12:**end for**


### 3.5. Analysis of the Algorithm’s Performance

To showcase the benefits of this algorithm, this study compares it with the clarity evaluation algorithm based on high-frequency sub-band coefficient energy. The comparison results are shown in Figure 3, where the vertical axis is the mapping of the sharpness evaluation value between 0 and 1, and the horizontal axis represents the row number of the pictures. For example, “5” indicates the fifth picture.

It can be seen from Figure 3 that the sharpness evaluation algorithm based on the NSST still shows strong judgment ability when there is only a small amount of image information. Compared with the algorithms that use high-frequency sub-band coefficient energy as the sharpness evaluation, the new algorithm has good unimodality and noise immunity, with all the advantages of spectral functions. However, due to the frequency domain decomposition of images into sub-images with different frequency bands in multiple directions, the running time of the algorithm will be slightly longer than that of the time-domain sharpness evaluation algorithm.

## 4. Results

The experimental environment: CPU—AMD Ryzen 7, 4800H, with Radeon Graphics, 2.90 GHz; RAM—16.00 GB; MATLAB (R2019b).

### 4.1. Analyses and Comparison of Algorithms

In order to make the experiment more rigorous and the algorithm more effective, 10 groups of defocus–focus–defocus urinary sediment cell images with 800 × 600 pixels are taken as test samples under the condition of 40X microscope amplification, with the object distance adjusted. Each group contains 19 to 22 images. In this study, four sets of experimental images and results are presented, and 10 of them are listed, as shown in Figure 4. The sharpness evaluation value of each image is calculated in MATLAB and plotted into a sharpness evaluation function curve. In addition, in order to analyze and view the results more conveniently and intuitively, this study normalizes all the results, as shown in Figure 5. The non-subsampled contourlet transform (NSCT), Tenengrad algorithm [29], Roberts algorithm [30,31], discrete cosine transform (DCT) [32,33], energy of gradient (EOG) [34] algorithm, Canny algorithm [35,36], and Laplacian algorithm [37,38] are selected for this comparative experiment. The Tenengrad algorithm uses a Sobel [39,40] operator to extract gradient values in horizontal and vertical directions, and then calculates the gradient square sum of all pixels. In addition, since the datasets involved in the experiment are small in size, the deep learning method cannot be used for comparative experiments. Therefore, no introduction will be made to deep learning in this study.

In order to verify that the algorithm is feasible on other types of images, two sets of public data and two sets of biomedical cell images with different contents are selected for testing. The experimental images and results are shown in Figure 6. Furthermore, Redondo et al. [41] defined the ratio of the width of the evaluation curve at 40% and 80% as a narrow width α/β, which can objectively reflect the performance of the algorithm. The values of the narrow width of the algorithms in the urinary sediment cell dataset, the public dataset, and biomedical cell images are shown in Table 1 and Table 2 below. The average time consumed by each algorithm is given in the tables.

Both advantages and disadvantages of an algorithm can be analyzed based on whether its sharpness evaluation curve has a single peak, whether the curve is steep, and whether there are local peaks and how high the local peaks are. It can be concluded from Figure 5 that in the four sets of comparison experiments, the algorithm proposed in this study can deliver steeper curves, lower local peaks, and higher sensitivity for defocus and in-focus image recognition than other algorithms. Specifically, the normalized evaluation value of the defocus images is close to 0, and that of the focused ones is close to 1, indicating that the algorithm proposed by this study is better than other algorithms reviewed by this study.

As shown in Table 1, the narrow width can objectively reflect the steepness of the sharpness evaluation curve: the larger the narrow width, the steeper the curve—in other words, the better the algorithm’s performance. Among the 11 groups in the test data, the narrow width in the improved NSST algorithm is the highest in 8 groups, and the second highest in 2 groups; besides, the narrow width can be about 100% higher than the lowest value and about 10–20% higher than the second highest value, also indicating that the improved NSST algorithm is superior to other algorithms in terms of the steepness of the sharpness evaluation curve, i.e., its curve is narrower. In the other two sets of data, the narrow width of the improved NSST algorithm delivers the highest or second highest value, about 5% higher than the second highest value in the ball dataset. On another set of biomedical image datasets, it can be clearly seen that the improved NSST algorithm has a good effect in processing such images. From the narrow width obtained, the improved NSST algorithm is 60-80% higher than other algorithms mentioned in this study. This demonstrates that the improved NSST algorithm is also appropriate for other microscope images, but with an effect not as good as that for urine sediment images.

### 4.2. Noise Immunity Test

While obtaining an image, microscopes often produce some noise due to environment and equipment problems. In order to precisely analyze the advantages and disadvantages of the experimental algorithm as well as its noise immunity performance, this study adds Gaussian, salt-and-pepper, and Poisson noise. After the noises are added, the images are processed via a bilateral filter and guided filter in order to enhance the anti-noise performance of the algorithm. The images with noises are shown in Figure 7, and the results have been normalized. The normalized curves are shown in Figure 8. The images’ normalization curves for the public dataset are shown in Figure 9. The images’ normalization curves for the biomedical cell image sets are shown in Figure 10.

It can be observed in the figures above that when the image is added with salt-and pepper-noise of a noise density value of 0.1, there will be large fluctuations in the clarity evaluation curve for the existing algorithm, multiple local maximums will appear, and the image cannot be intuitively reflected. In other words, it cannot reflect unimodality and unbiasedness. In addition, the principle of the NSCT algorithm is similar to the NSST algorithm, and thus, NSCT can provide better noise immunity performance for the existing algorithms, but it is weaker than the NSST algorithm as mentioned above. When Gaussian noise with a noise density value of 0.1 is added to the image, the existing algorithms can find the global maximum, but there are still fluctuations in the sharpness curve; in contrast, the improved NSST algorithm proposed by this study does not have such problems. In addition, the narrow width α/β can determine the steepness of the curve. After noises are added, all the existing algorithms showcase upward or downward fluctuations to varying degrees. Therefore, it is impossible or difficult to find the narrow width. However, the improved NSST used in this study would not encounter this problem, demonstrating that the anti-noise performance of the improved NSST algorithm adopted in this study performs better than other algorithms reviewed by this study.

## 5. Conclusions

This study proposes an improved sharpness evaluation algorithm based on the improved NSST for the sharpness evaluation of microscope images, and compared it with other algorithms reviewed by this study. It is concluded that the other algorithms in the sharpness evaluation of microscope images are less feasible than the algorithm proposed in this study. Other algorithms compared in this study cannot well reflect the unimodality and noise resistance after different noises are added. The improved NSST algorithm can perfectly reflect these two points. Additionally, the improved NSST algorithm is especially suitable for the urine sediment cell images, because the background of such images is monotonous, the texture of the cell object is remarkable, and the degree of color interference is weak. In addition, the improved NSST delivers better results than other methods referred to in this study. However, the improved NSST algorithm still has certain shortcomings for noise immunity. For example, when the noise intensity is strong to a certain degree, the sharpness evaluation curve of the improved NSST will inevitably showcase upward and downward fluctuations. In addition, because the improved NSST performs multidirectional subtraction in the frequency domain, the complex decomposition of bands will cause a slightly longer calculation time than the time-domain clarity evaluation algorithm. Since this study aims to find an algorithm with good noise immunity and sensitivity, the impact of time consumption is ignored in this study. However, in the future, it is necessary to conduct more in-depth mining and research on the optimization of the algorithm’s running time and anti-noise performance.

## Figures and Tables

**Figure 1 sensors-22-07607-f001:**
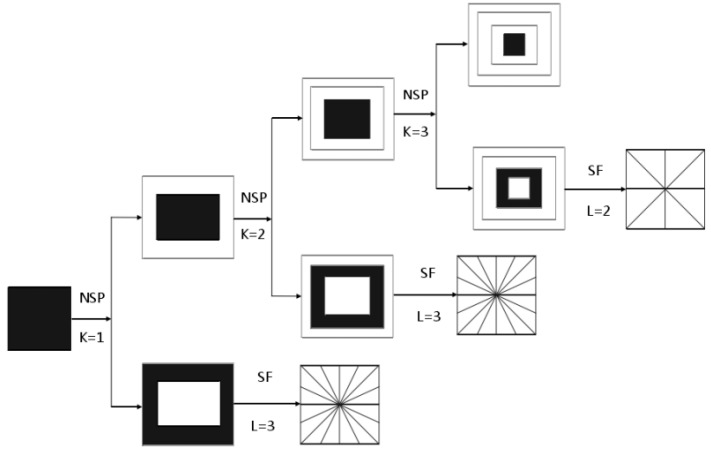
NSST breakdown structure diagram.

**Figure 2 sensors-22-07607-f002:**
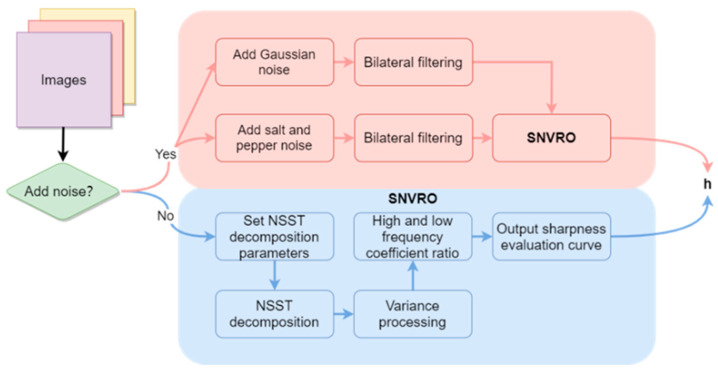
Flow chart of sharpness evaluation curve.

**Figure 3 sensors-22-07607-f003:**
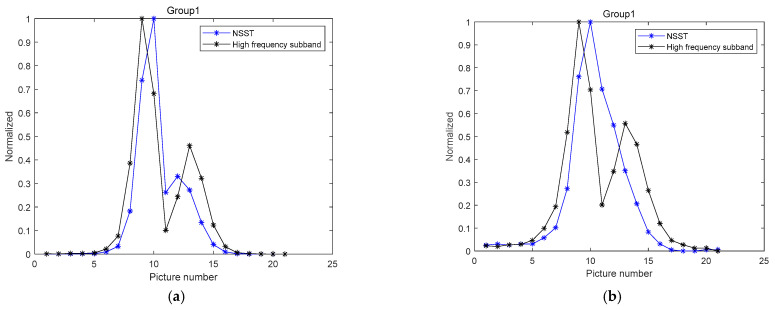
Comparison results of algorithm performance with and without noise: (**a**) image without noise; (**b**) image with noise.

**Figure 4 sensors-22-07607-f004:**
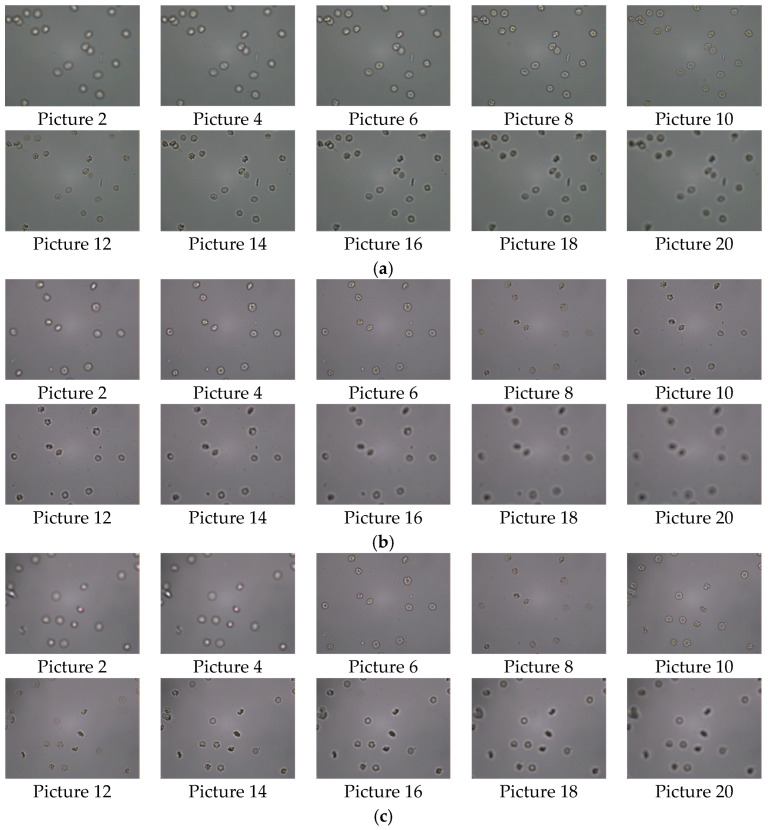
Four groups of test experiment diagrams: (**a**) Group 1; (**b**) Group 3; (**c**) Group 4; (**d**) Group 9.

**Figure 5 sensors-22-07607-f005:**
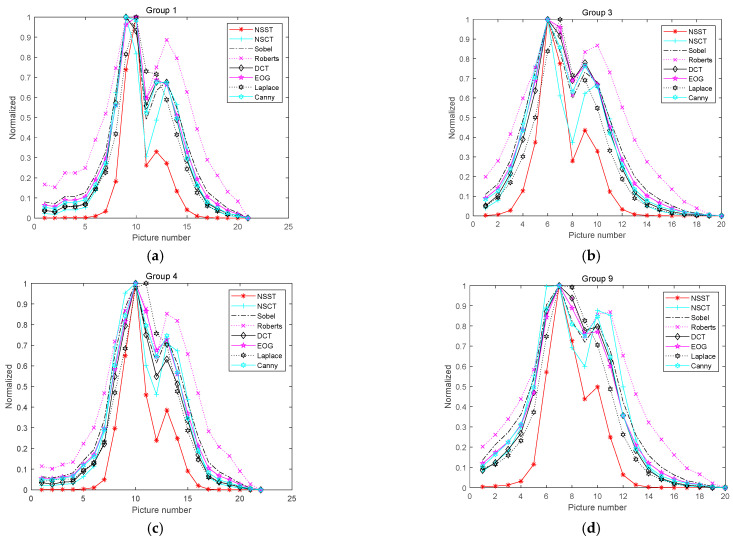
Sharpness evaluation curve of each algorithm for the four groups of images: (**a**) Group 1; (**b**) Group 3; (**c**) Group 4; (**d**) Group 9.

**Figure 6 sensors-22-07607-f006:**
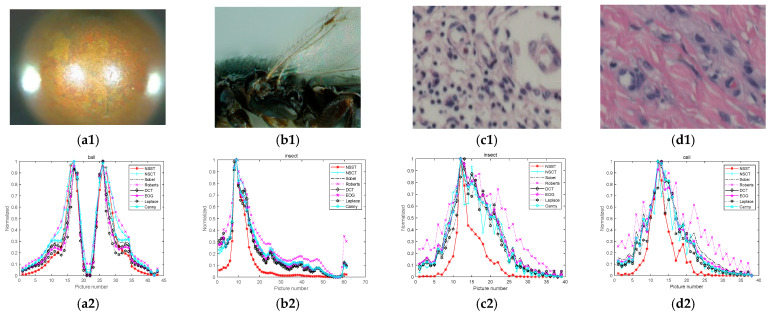
Sharpness evaluation curve of each algorithm for the public data: (**a1**) ball image; (**b1**) insect image; (**c1**,**d1**) biomedical cell images; (**a2**) ball sharpness evaluation curve; (**b2**) insect sharpness evaluation curve; (**c2**,**d2**) biomedical cell image sharpness evaluation curve.

**Figure 7 sensors-22-07607-f007:**
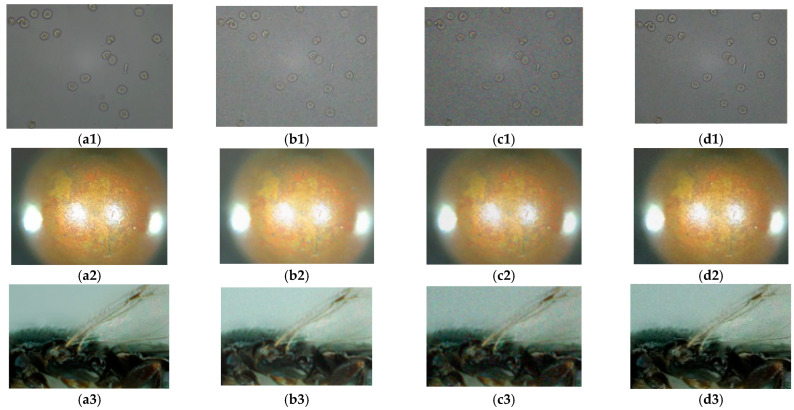
Image comparison: (**a1**,**b1**,**c1**,**d1**) are the datasets used in this study; (**a2**)–(**d5**) are two public datasets and two biomedical cell image sets; (**a1**–**a5**) clear cell image; (**b1**–**b5**) images with Gaussian noise added; (**c1**–**c5**) images with salt-and-pepper noise; (**d1**–**d5**) images with Poisson noise.

**Figure 8 sensors-22-07607-f008:**
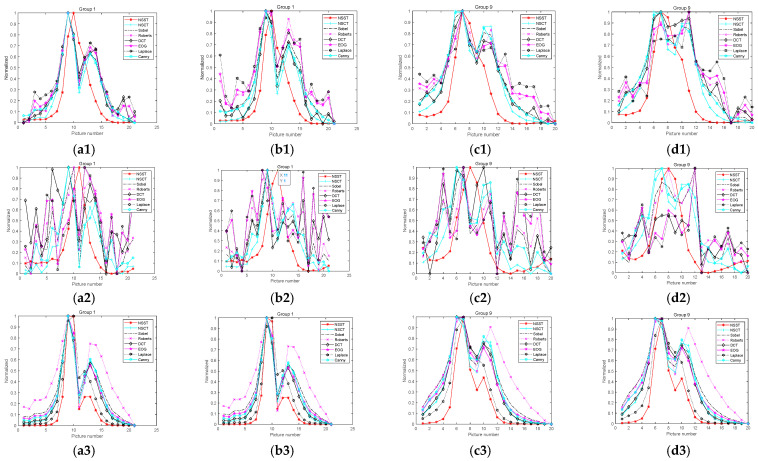
Curves after noises and filters are added: (**a1**) Gaussian noise and bilateral filter of Group 1; (**b1**) Gaussian noise and guided filter of Group 1; (**c1**) Gaussian noise and bilateral filter of Group 9; (**d1**) Gaussian noise and guided filter of Group 9; (**a2**) salt-and-pepper noise and bilateral filter of Group 1; (**b2**) salt-and-pepper noise and guided filter of Group 1; (**c2**) salt-and-pepper noise and bilateral filter of Group 9; (**d2**) salt-and-pepper noise and guided filter of Group 9; (**a3**) Poisson noise and bilateral filter of Group 1; (**b3**) Poisson noise and guided filter of Group 1; (**c3**) Poisson noise and bilateral filter of Group 9; (**d3**) Poisson noise and guided filter of Group 9.

**Figure 9 sensors-22-07607-f009:**
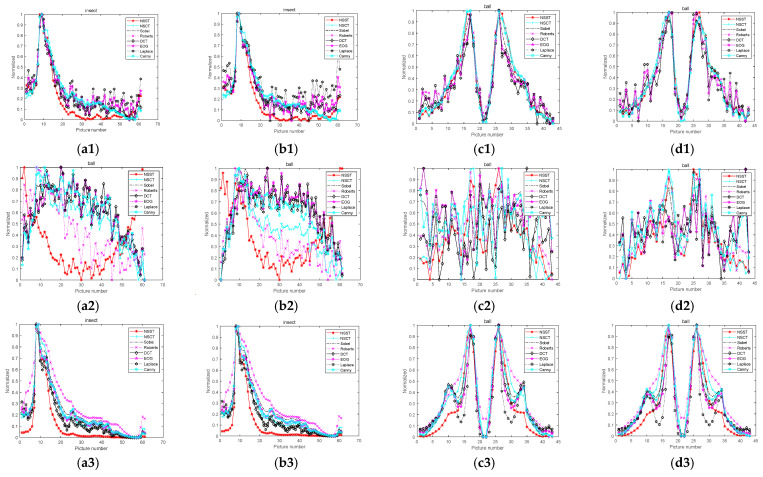
Curves after noises and filters are added: (**a1**) Gaussian noise and bilateral filter of insects; (**b1**) Gaussian noise and guided filter of insects; (**c1**) Gaussian noise and bilateral filter of ball; (**d1**) Gaussian noise and guided filter of ball; (**a2**) salt-and-pepper noise and bilateral filter of insects; (**b2**) salt-and-pepper noise and guided filter of insects; (**c2**) salt-and-pepper noise and bilateral filter of ball; (**d2**) salt-and-pepper noise and guided filter of ball; (**a3**) Poisson noise and bilateral filter of insects; (**b3**) Poisson noise and guided filter of insects; (**c3**) Poisson noise and bilateral filter of ball; (**d3**) Poisson noise and guided filter of ball.

**Figure 10 sensors-22-07607-f010:**
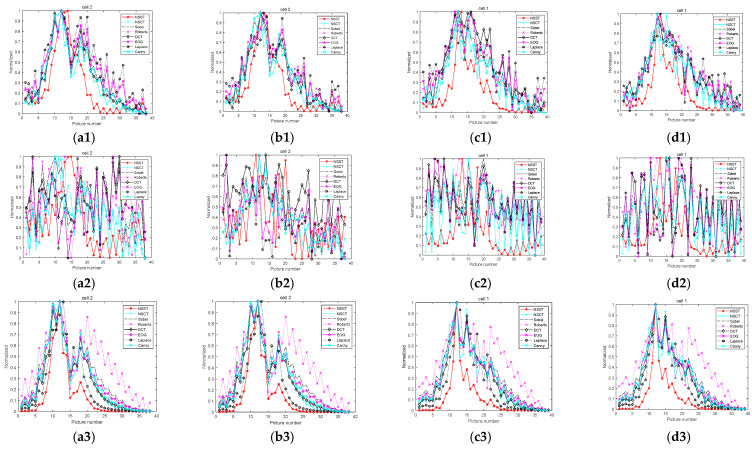
Curves after noises and filters are added: (**a1**) Gaussian noise and bilateral filter of cell 2; (**b1**) Gaussian noise and guided filter of cell 2; (**c1**) Gaussian noise and bilateral filter of cell 1; (**d1**) Gaussian noise and guided filter of cell 1; (**a2**) salt-and-pepper noise and bilateral filter of cell 2; (**b2**) salt-and-pepper noise and guided filter of cell 2; (**c2**) salt-and-pepper noise and bilateral filter of cell 1; (**d2**) salt-and-pepper noise and guided filter of cell 1; (**a3**) Poisson noise and bilateral filter of cell 2; (**b3**) Poisson noise and guided filter of cell 2; (**c3**) Poisson noise and bilateral filter of cell 1; (**d3**) Poisson noise and guided filter of cell 1.

**Table 1 sensors-22-07607-t001:** Comparison of narrow widths of noise-free images in different algorithms. The maximum value is given in bold.

Algorithms	1	2	3	4	5	6	7	8	9	10	11	Time
NSCT	**0.4545**	0.3771	0.3238	0.2672	0.4120	0.3974	0.4322	0.3541	0.2642	0.4577	0.3778	135.54
Sobel	0.2433	0.2699	0.2398	0.2853	0.3552	0.2696	0.3659	0.1550	0.3197	0.2915	0.2473	2.40
Roberts	0.2247	0.2254	0.2249	0.2861	0.2879	0.2823	0.2019	0.2009	0.2994	0.2366	0.2423	2.33
DCT	0.2631	0.2872	0.2906	0.2518	0.4054	0.2662	0.3717	0.1876	0.4235	0.2989	0.2759	10.98
EOG	0.2635	0.3048	0.2972	0.3224	0.3537	0.2431	0.1902	0.1790	0.3892	0.2819	0.2929	2.34
Laplacian	0.2874	0.3093	0.2959	**0.3639**	0.2859	0.2139	0.1614	0.2420	**0.4768**	0.2274	0.2666	2.34
Canny	0.2733	0.2981	0.3116	0.2766	0.3921	0.2625	0.2123	0.1875	0.2985	0.3092	0.2703	9.36
NSST	0.4258	**0.5170**	**0.4500**	0.3248	**0.4899**	**0.4224**	**0.4417**	**0.3587**	0.3478	**0.5029**	**0.5263**	33.97

**Table 2 sensors-22-07607-t002:** Comparison of images’ narrow widths of noise-free public dataset in different algorithms, where ball (left peak) represents the peak on the left and ball (right peak) represents the peak on the right. The maximum value is given in bold.

Algorithms	Ball (Left Peak)	Ball (Right Peak)	Insect	Cells	Time
NSCT	0.4318	0.4026	**0.4085**	0.1897	394.67
Sobel	0.4361	0.3454	0.2667	0.1304	5.19
Roberts	0.3586	0.2988	0.3245	0.2042	5.95
DCT	0.2412	0.2111	0.3131	0.1970	31.79
EOG	0.4473	0.3889	0.2441	0.2288	5.96
Laplacian	0.4492	0.4078	0.2586	0.2429	5.97
Canny	0.4393	0.3780	0.2738	0.2653	52.98
NSST	**0.4719**	**0.4196**	0.3956	**0.3146**	77.86

## Data Availability

Data available on request.

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
