# Peer review of "A Method for Medical Microscopic Images’ Sharpness Evaluation Based on NSST and Variance by Combining Time and Frequency Domains"

_sensors, 2022, doi:10.3390/s22197607_

Round 1
Reviewer 1 Report
The authors seem to have put their efforts to present their methodoloy. However, the subject is well known to the readers with an appropriate knowledge and interest about the methods of sharpness evaluation whilst the algorithms based on NSST for the sharpness evaluation of microscope images have good results and accuracy degree so far. The authors, hence, and for a better presentation of their achievements, should show their algorithm performance on images with rather more complicated noise, more natural type, not only salt and pepper or Gaussian noise. I suggest they should consider medical or rather biological images and check the results of applying other filters (Canny filter, for example) in addition to those mentioned and studied by the authors. I think this way they can enhance the methodology with something new and show how exactly their method outperforms the known ones.
Moreover, the authors should consider more recent references for comparison. There is no reference from 2022 whilst the 2 or 3 ones from 2021 are not of great significance for comparison.
Author Response
Please see the attachment
The modified part is highlighted in yellow

Reviewer 2 Report
#NSST (Non-subsampled shearlet wave transform) --> Non-subsampled shearlet wave transform(NSST)
#Clearly write motivation, contribution and novelty in the Introduction section.
#Add manuscript organization at the end of the Introduction section.
#Add a separate related work section and highlight the research gap.
#Authors must go through the work like Light microscopy image de-noising using optimized LPA-ICI filter. Neural Computing and Applications, Finding contours of hippocampus brain cell using microscopic image analysis. Journal of Advanced Microscopy Research, 
# Write the motivation of using NSTT.
#Add a discussion section. Discuss about Nonparametric de‐noising filter optimization using structure‐based microscopic image classification. Microscopy research and technique, Digital analysis of microscopic images in medicine. Journal of Advanced Microscopy Research , limitations scope etc.
#There are many references where there is an issue of capitalization.
#Provide DOI for all ref. or not any one.
#Report the time complexity.
Author Response
Please see the attachment, all changes are highlighted

Round 2
Reviewer 1 Report
The paper looks much better now. However, it will really be a very good work if the authors give medical examples instead of (or at least in addition to) the insect image. The authors mentioned other image examples in the introduction. Hence, they can simpley use them as examples. No special need to give further details but show how your algorithm works on medical images.
Author Response

(The authors gave the same response as above.)

Reviewer 2 Report
Well revised
Author Response
Dear Reviewer,
Thank you very much for your time involved in reviewing the manuscript and making valuable comments